# *DEHA*-Net: A Dual-Encoder-Based Hard Attention Network with an Adaptive ROI Mechanism for Lung Nodule Segmentation

**DOI:** 10.3390/s23041989

**Published:** 2023-02-10

**Authors:** Muhammad Usman, Yeong-Gil Shin

**Affiliations:** Department of Computer Science and Engineering, Seoul National University, 1 Gwanak-ro, Gwanak-gu, Seoul 08826, Republic of Korea

**Keywords:** lung nodule segmentation, 3D segmentation, dual-encoder-based CNN, hard attention

## Abstract

Measuring pulmonary nodules accurately can help the early diagnosis of lung cancer, which can increase the survival rate among patients. Numerous techniques for lung nodule segmentation have been developed; however, most of them either rely on the 3D volumetric region of interest (VOI) input by radiologists or use the 2D fixed region of interest (ROI) for all the slices of computed tomography (CT) scan. These methods only consider the presence of nodules within the given VOI, which limits the networks’ ability to detect nodules outside the VOI and can also encompass unnecessary structures in the VOI, leading to potentially inaccurate segmentation. In this work, we propose a novel approach for 3D lung nodule segmentation that utilizes the 2D region of interest (ROI) inputted from a radiologist or computer-aided detection (CADe) system. Concretely, we developed a two-stage lung nodule segmentation technique. Firstly, we designed a dual-encoder-based hard attention network (*DEHA*-Net) in which the full axial slice of thoracic computed tomography (CT) scan, along with an ROI mask, were considered as input to segment the lung nodule in the given slice. The output of *DEHA*-Net, the segmentation mask of the lung nodule, was inputted to the adaptive region of interest (A-ROI) algorithm to automatically generate the ROI masks for the surrounding slices, which eliminated the need for any further inputs from radiologists. After extracting the segmentation along the axial axis, at the second stage, we further investigated the lung nodule along sagittal and coronal views by employing *DEHA*-Net. All the estimated masks were inputted into the consensus module to obtain the final volumetric segmentation of the nodule. The proposed scheme was rigorously evaluated on the lung image database consortium and image database resource initiative (LIDC/IDRI) dataset, and an extensive analysis of the results was performed. The quantitative analysis showed that the proposed method not only improved the existing state-of-the-art methods in terms of dice score but also showed significant robustness against different types, shapes, and dimensions of the lung nodules. The proposed framework achieved the average dice score, sensitivity, and positive predictive value of 87.91%, 90.84%, and 89.56%, respectively.

## 1. Introduction

Lung cancer is the deadliest cancer type, and early detection is crucial for potentially life-saving treatment. The accurate quantification of pulmonary nodules, which may be associated with various conditions but are often indicative of lung cancer, is essential for the continuous monitoring of lung nodule volume to assess the malignancy and predict the likelihood of lung cancer [1,2]. However, the manual segmentation of nodules, which represents a necessary step in calculating their volume, is a laborious and time-consuming process that can also introduce variability between and within observers [3].

Computer-aided diagnosis (CAD) systems can significantly enhance the productivity of radiologists by assisting them in overcoming the challenges associated with the manual segmentation of pulmonary nodules. CAD systems consist of two subsystems, i.e., computer-aided detection (CADe) [4] and computer-aided diagnosis (CADx) [5]. The CADe system aims to distinguish between nodules and other structures, such as tissues and blood vessels. The CADx system then evaluates the detected nodules and determines whether they are benign or malignant tumors. The primary goal of these CAD systems is to improve the accuracy and efficiency of cancer diagnosis by radiologists. They are designed to assist decision making by providing additional information and reducing the time needed to interpret CT images. This work is focused on developing the CADx system for accurate lung nodule segmentation, which is challenging due to variable shapes, different sizes, and complicated surrounding tissues in the lung region. Various automatic segmentation frameworks for nodule quantification have been devised; such techniques consist of traditional image-processing-based methods and deep-learning-based approaches [6]. However, the significant heterogeneity of lung nodules, particularly the variations in the shape and contrast of lung nodules, hinders the development of a robust nodule segmentation framework. These variations, both within and between nodules, as well as the visual similarity between nodules and their surrounding non-nodule tissue, necessitate the use of a 3D volume of interest (VOI) as input to estimate the shape of the nodule accurately. Figure 1 demonstrates the intra-nodule and inter-nodule variations, showcasing the diversity between the forms of different nodules and the variations present within individual nodules. Providing a 3D VOI is quite a time-consuming and laborious task, as the radiologist has to specify the region of interest at each slice containing the nodule. A few studies have resolved this issue by utilizing a fixed ROI for all the slices; this approach requires only one ROI input from the user, which significantly reduces the time and hassle. However, employing a fixed ROI adds redundant non-nodular regions to the input ROI, leading to poor segmentation performance.

To address the issues related to using 3D VOIs as input and fixed ROIs, in our previous work [5], we proposed a novel approach using dynamic ROIs for the accurate volumetric segmentation of pulmonary nodules. To determine the dynamic ROIs, we proposed an adaptive region of interest (A-ROI) algorithm that utilizes a single 2D ROI provided by radiologists [5] to estimate the dynamic ROIs in the surrounding slices. This approach begins by segmenting the nodule in the initially provided ROI by employing residual-UNet and then utilizes the segmentation mask to determine the ROIs for the surrounding slices to extend the nodule segmentation in both directions. Concretely, the A-ROI algorithm dynamically adjusts the position and size of the bounding box for the adjacent slices to investigate the penetration of the nodule in the other slices. The technique demonstrated exceptional performance and outperformed the previous state-of-the-art methods. However, this previous approach required cropping the ROI, which can cause problems due to the inconsistent size of nodules, for instance, if the ROIs are too small or larger than the normalized dimensions used to input into the network. Similarly, the mask obtained after inference must be resized to match the original cropped ROI size, introducing error when interpolation is used to achieve the target dimensions. To address these issues, we propose a dual-encoder-based architecture that takes two inputs: the original slices and the ROI mask, eliminating the need to rescale the ROIs before and after inference. The A-ROI algorithm was then used to further produce ROI masks for surrounding slices for which ROIs were not provided. Specifically, the A-ROI algorithm was applied along the axial plane to provide an initial estimation of nodule shape, which was then used to extract a 3D VOI from the scan automatically. The extracted VOI was further utilized to create the coronal and sagittal views of the nodule, and the slices from these views were analyzed using two different dual-encoder-based architectures. Finally, a consensus module was employed to ensemble the three predictions from axial, coronal, and sagittal view models. Several experiments were performed on the LIDC dataset [7] to demonstrate the effectiveness of the proposed technique in terms of overall performance and robustness relative to the variations in the type and size of lung nodules.

## 2. Related Work

An accurate assessment of lung nodules is essential for evaluating their potential malignancy and the likelihood of being indicative of lung cancer. Subsequently, numerous researchers have made extensive efforts to devise an efficient nodule segmentation framework to assist radiologists. These studies can be classified into two categories, i.e., conventional image-processing-based methods and advanced deep-learning-based techniques [6].

Jamshid et al. [8] proposed a framework that segmented the nodule by employing region-growing techniques, such as contrast-based region growing and fuzzy connectivity region growing, and created a volumetric mask using a local adaptive segmentation algorithm that distinguishes between foreground and background regions within a specified window size. While the algorithm demonstrated good performance for isolated nodules, it could not effectively segment the attached ones. Using geodesic impact zones in a multi-threshold picture representation, Stefano et al. [9] offered a user-interactive algorithm that meets the fusion-segregation criterion based on both gray-level similarity and object shape. They extended their work in another study [10] by eliminating the need for user interaction. A correction procedure was then performed based on a 3D local shape analysis, allowing for the refinement of an initial nodule segmentation to distinguish possible vessels from the nodule itself without requiring input from the user. Rendon et al. [11] used morphological and threshold approaches to eliminate extraneous structures from a given ROI. The last step was to use a support vector machine (SVM) to categorize each pixel in the discovered space.

Although classical image-processing-based techniques achieve accurate lung nodule segmentation, such techniques are susceptible to the types of nodules. In contrast, recent deep-learning-based methods have made wast inroads into many medical imaging applications such as disease classification [12] and segmentation applications [13,14], including lung nodule segmentation [15]. The introduction of the UNet [16] architecture for medical image segmentation, in particular, has dramatically enhanced the performance of various crucial tasks, such as tumor segmentation [17]. As a result, there has been an increased focus on using deep learning for lung nodule segmentation. In [18], Tyagi et al. proposed a 3D conditional generative adversarial network (GAN) for lung nodule segmentation. They utilized the UNet architecture as the backbone of GAN. They employed a simple classification network as a discriminator, incorporating spatial squeeze, and channel excitation modules to differentiate between truth and fake segmentation. Similarly, Wang et al. [19] developed a method for nodule segmentation called central-focused convolutional neural networks (CF-CNNs). This approach uses a volumetric patch centered around the voxel of interest as input to the model. In addition, the team [20] also published a multi-view CNN that can perform nodule segmentation using input from different views (axial, coronal, and sagittal) of the same voxel. One potential limitation of this method is that the patch extraction process is the same for all nodules, which could lead to incorrect segmentation if the nodule is larger than the size of the patch. By using skip connections in the encoder and decoder paths, Tong et al. [21] enhanced the performance of UNet for nodule segmentation; however, the model was only intended for 2D segmentation. Hancock et al. [22] put forth a variation on the standard level-set picture segmentation technique in which, as opposed to being manually created, the velocity function is instead learned from data using regression machine learning techniques. They reported slightly improved performance when they applied this segmentation approach to the segmentation of lung nodules. Chen et al. [23] proposed an end-to-end multi-task learning framework that consists of joint classification and multi-channel segmentation networks. Both networks utilized the exact latent representation learned by the common encoder branch, improving lung segmentation performance. The study also incorporated an enhanced version of patches by using OTSU and SLIC methods. To extract local characteristics and detailed contextual information from lung nodules, Liu et al. [24] used a residual-block-based dual-path network, which significantly improved performance. They also employed a fixed VOI, which restricts the nodule search and lowers 3D segmentation performance. To avoid this issue, Chen et al. [25] proposed a fast multi-crop guided attention (FMGA) network for lung nodule segmentation by incorporating 2D- and 3D-cropped ROIs. They applied the greedy search algorithm to explore the penetration of lung nodules into the surrounding slices. Their framework also exploited a customized loss function, enabling the network to focus on improving the segmentation of nodule borders. Their results demonstrated the robustness of the proposed framework; however, the scheme failed to improve state-of-the-art methods in terms of the overall dice score.

In our previous work [5], we addressed the limitations of a fixed volume of interest (VOI) by introducing the concept of an adaptive 2D region of interest (ROI) in each slice, which significantly improved the ability to utilize deep learning. Most notably, cropped ROIs were fed to the deep residual UNet [26], which demonstrated promising performance along with several limitations. Particularly, due to the heterogeneity of lung nodules, numerous variations in dimensions are possible, which makes it impossible to find the optimal input dimensions for the network. Subsequently, the cropped ROI has to be severally resized, by upsampling or downsampling the ROI, which affects the performance of the proposed framework. One possible alternative is to train various models with different input dimensions. However, this comes with an immense increase in the computational cost, which hinders the solution’s real-time clinical applications. For instance, Zhang et al. [27] proposed multi-scale segmentation squeeze-and-excitation UNet with a conditional random field to segment the nodule in the given volume of interest. They extracted VOIs at four different scales and trained four different networks and finally applied a conditional random field to merge the four predictions. Their framework increased the computational complexity and only covered four scales defined according to dimensions available in the given dataset, which is insufficient to cover the possible diversity in the size of lung nodules in real-time clinical applications. To overcome the aforementioned issue, in this work, we propose a dual-encoder-based architecture that incorporates the ROI mask to input as hard attention, which enables the framework to avoid the pre- and post-inference resizing and leads to performance improvement.

## 3. Materials and Methods

### 3.1. Dataset

In this work, we used the lung image database consortium and image database resource initiative (LIDC-IDRI) database [7,28], which is the largest publicly available resource for lung CT scans. This dataset is created to facilitate the development of computer-aided systems for evaluating lung nodule identification, categorization, and quantification. In the LIDC-IDRI, a sizable number of thoracic CT scans have been gathered; the database comprises 1018 diagnostics and screening thoracic CT images for lung cancer from 1010 individuals with annotated lesions. Each thoracic CT scan underwent a two-phase annotation process performed by four qualified radiologists. As in earlier studies [5,29,30], in this work, we also considered nodules with a minimum diameter of 3 mm and annotations from all four radiologists. The ground-truth border for pulmonary nodule segmentation was created using a 50% consensus criterion [31] due to the variability among the four radiologists, and a Python module named pyLIDC was employed. A total of 893 nodules from the LIDC dataset were selected and randomly distributed into 40%, 5%, and 55% sets, which were, respectively, used as training, validation, and test sets.

### 3.2. Data Pre-Processing

The pre-possessing of CT images can significantly improve the network’s performance by reducing the influence of noise and irrelevant tissues. Normalizing the image can reduce the network’s dependence on parameter initialization, smoothing the optimization process, and, subsequently, enhancing the convergence probability. Concretely, grayscale thresholding was applied to normalize the intensity range, which helped to suppress irrelevant, redundant information. This enabled the network to pay attention to the relevant tissue and reduce the complexity of the input data, making the network’s training more efficient and effective.

We also normalized the intensity values, ranging from 0 to 1, by using the window center and window width tag from corresponding DICOM files [32]. The normalization can be defined as follows:(1)$$I_{n}=\frac{I-W_{Min}}{W_{Max}-W_{Min}},$$
(2)$$W_{Min}=WC-WW/2,$$
(3)$$W_{Max}=WC+WW/2,$$
where *I*, $I_{n}$, WC, and WW represent the original image, normalized image, window center, and window width, respectively. The values of the window center and window width are extracted from the DICOM tags [32].

The LIDC collection includes the scans obtained from numerous locations and scanners. Consequently, it has a variety of pixel spacings and slice thicknesses. These variables are crucial for nodule appearance. In particular, slice thickness significantly impacts the coronal and sagittal views. Slice thickness in most LIDC scans, which spans from 0.45 mm to 5.0 mm, is higher than pixel spacing. Therefore, to enhance the visibility of nodules in all three views, the slice thickness was reduced to the corresponding pixel spacing by upsampling the scan along the z-axis. The pixel spacing remained unchanged, as it was less than one for each scan, producing an axial view of the nodules in a reasonable resolution.

In contrast to previous studies [19,20,21,33], which produced the training samples by employing the constant margin scheme, in this work, we utilized the ROI with random margins on each side as in [5]. To train our *DEHA*-Net architecture, we generated ROI masks by using ground-truth nodule masks. To enforce our model to learn about the absence of lung nodules in a given slice, we also included two non-nodular slices from both sides of each nodule.

### 3.3. Dual-Encoder-Based Hard Attention Network with Adaptive ROI Mechanism

The proposed framework utilized a novel dual-Encoder-based hard attention network (*DEHA*-Net) with an adaptive ROI (A-ROI) mechanism. The overall framework is illustrated in Figure 2. The first stage, the 2D ROI, which used manual input from a radiologist or computer-aided diagnosis (CADe) system as its source, was carried out using *DEHA*-Net along the axial axis. The A-ROI algorithm was applied to generate the ROIs for the remaining surrounding slices, which enabled the investigation of the nodules along the axial view in order to reconstruct the 3D mask of the nodule. In the second stage, the 3D mask constructed after the axial analysis was exploited to generate the ROIs along the sagittal and coronal views. Then, we applied the proposed *DEHA*-Net along the sagittal and coronal views with predefined ROIs generated from the 3D mask obtained at the first stage. Finally, a consensus module was utilized to produce the final 3D segmentation mask of the nodule. It is important to note that in the whole pipeline, no resizing was performed, thus eliminating the issues associated with the rescaling of a given input and network output. This enabled our network to achieve improved performance and made it more robust to size variations in various nodules. The following subsection describes the details of the proposed *DEHA*-Net and the A-ROI algorithm.

#### Dual-Encoder-Based Hard Attention Network

Lung nodules vary in shape and dimension, making it impossible to set a suitable input dimension for the network. To overcome this issue, we designed a dual-encoder-based hard attention network (*DEHA*-Net) that incorporated two inputs, i.e., the slice containing the nodule and ROI mask, to segment the nodule in the given slice accurately. Specifically, the ROI mask provided hard attention, which enabled the network to focus on only the provided region of interest. The proposed *DEHA*-Net consisted of two encoders and one decoder branch, as demonstrated in Figure 3. Each encoder was connected to the decoder with residual connections from four different levels. Both encoders had identical architecture, consisting of four levels. At *nth* level, there was a convolution layer of $32\times n^{2}$ filters and a kernel size of 3 × 3, followed by rectified linear activation to add non-linearity. After ReLU, there was a batch normalization layer and then a max-pooling layer, which compressed the information. These four layers made a single level of an encoder.

The first encoder extracted the features from CT scan images, while the second encoder enforced the hard attention learned from the ROI mask of the nodule. Its primary purpose was to maintain the network’s focus on the nodule’s location. Decoders output the segmentation mask of nodule for the current mask and ROI for the next and previous slice. Similarly, the decoder consisted of four levels, each consisting of a concatenation layer followed by a convolution layer. After that, rectified linear was applied for non-linearity, followed by a batch normalization layer, and finally, these features were upsampled. In the last level of the decoder, the upsampling layer was replaced by a convolution layer of a single filter with SoftMax activation. Each concatenation layer of the decoder concatenated the features from each level of both encoders and the previous decoder level to pass into the proceeding layers.

### 3.4. Adaptive ROI Algorithm

The adaptive ROI (A-ROI) algorithm was proposed in [5], which enables the network to investigate nodule presence in the surrounding slices without having ROIs from the user. Concretely, the A-ROI algorithm exploits the segmented mask of nodules in the current (*nth*) slice generated by the network to estimate the ROI for the successive slices (i.e., n±1). In this work, we employed the A-ROI algorithm to complement the proposed *DEHA*-Net to perform the 3D segmentation of lung nodules. A-ROI utilizes a hyperparameter $R_{T}\in \left(0,1\right)$ to moderate the margins around the nodule in the generated ROI masks.

The full impact of the A-ROI algorithm is demonstrated in Figure 4. Two constant ROIs are shown in the first row in red and blue, which remain fixed throughout all the slices: One with tight margins failed to cover the nodule in the surrounding slices, while the other constant ROI had wider margins, which added redundant area, thus confusing the network. By contrast, in the second row, the dynamic ROI produced by the A-ROI algorithm is shown. The column shows the different slices; (a) represents the slice where the user provides the first ROI, and (b–f) demonstrate the adjacent slices.

The proposed framework for generating the 3D segmentation mask of lung nodules along the axial view is described in Algorithm 1. Algorithm 1 illustrates the steps followed to generate the 3D segmentation mask by investigating nodule penetration along the axial axis. The provided ROI by the radiologist or CADe system in $n_{i}th$ slice is represented by $RoI_{n_{i}}$ and is used as $RoI_{n}$ to initiate the segmentation. The normalized slice, $I_{n}$, and the provided ROI are fed to *DEHA*-Net, denoted by Θ. Later, the segmentation mask of the nodule generated by *DEHA*-Net was inputted into the A-ROI algorithm to produce the ROI mask for the next slice. The next slice could be in any direction, i.e., forward or backward. The same cycle was repeated until the next ROI mask became blank.
**Algorithm 1**: The algorithmic steps followed in the proposed framework for nodule investigation along the axial view.1:$n=n_{i}$, $RoI_{n}=RoI_{n_{i}}$2:**while** 
$\sum RoI_{n}>0$
 **do**3:    $Seg_{n}=\Theta \left(I_{n},RoI_{n}\right)$4:    n←n±15:    $RoI_{n}\leftarrow AROI\left(Seg_{n},R_{T}\right)$6:**end while**

### 3.5. Ensembling Mechanism

The proposed framework utilized a consensus module to ensemble the segmentation results obtained from the axial, sagittal, and coronal axes. The consensus value $E_{i}$ of $i_{th}$ voxel is calculated as follows:(4)$$E_{i}=thr\left(\sum_{j=1}^{K}S_{ij},\tau \right)$$
(5)$$thr\left(g,\tau \right)=\left\{\begin{array}{cc} 1, & if\;g\geq \tau \\ 0, & Otherwise \end{array}\right.$$
where $S_{ij}$ represents the prediction of ith from the jth model, and *K* denotes the number of models, which in our case were three, i.e., axial, sagittal, and coronal. τ is the threshold that is determined on the validation set.

## 4. Experimental Setup and Implementation Details

### 4.1. Loss Function

To train the proposed *DEHA*-Net, we utilized the dice similarity coefficient (DSC) [34] loss, which can be defined as follows:(6)$$L_{\mathrm{DSC}}=\frac{1}{N}\sum_{i=1}^{N}\left[1-\frac{2*\Theta \left(I_{i},RoI_{i}\right)\cap S_{g_{i}}}{\Theta \left(I_{i},RoI_{i}\right)+S_{g_{i}}}\right]$$
where Θ, $S_{g_{i}}$, and *N* represent the model, ground-truth segmentation mask, and the number of samples in the training set, respectively. We used stochastic gradient descent (SGD) to train our network.

### 4.2. Implementation Details and Training Strategy

We used the Keras [35] framework for implementing the proposed *DEHA*-Net and used Equation (Equation 6) with an SGD scheme to minimize the error. The model was trained on an Nvidia Tesla V100 Tensor core GPU with 12,821 images sized 512×512 and a batch size of 8. Training was initiated from random weights and with an initial learning rate of 0.001 and the first and second momentum of 0.9 for the decay of the learning rate. We used early stopping with a patience value of 10 epochs to avoid overfitting.

### 4.3. Performance Measures

We considered three evaluation parameters to rigorously evaluate the performance of the proposed framework. The following evaluation parameters were used to evaluate the performance of our proposed method.

**Dice Similarity Coefficient:** We used the dice similarity coefficient (*DSC*) [19,36], which measures the degree of overlap between the ground-truth mask and the predicted mask. The *DSC* values range from 0 to 1, where 1 and 0 indicate complete overlap and no overlap, respectively. It can be defined as follows:
(7)$$DSC=\frac{2*Y^{\prime }\cap Y}{Y^{\prime }\;\cup \;Y}.$$
where $Y^{\prime }$ and *Y* are the predicted segmentation mask and reference segment mask, respectively.**Sensitivity:** To measure the pixel classification performance proposed framework, we used sensitivity (*SEN*), which can be defined as follows:
(8)$$SEN=\frac{Y^{\prime }\cap Y}{Y}.$$**Positive Predictive Value (*PPV*):** To measure the correctness of the segmentation area produced by the proposed framework, we used the positive predictive value (*PPV*), which can be defined as follows:
(9)$$PPV=\frac{Y^{\prime }\cap Y}{Y^{\prime }}.$$

## 5. Results and Discussion

### 5.1. Overall Performance Analysis

We evaluated our proposed framework on the parameters described in Section 4.3 and compared its performance with previously published studies. Table 1 summarizes the results achieved by using our framework on the test set along with the reported performance of existing studies. It demonstrates that our proposed architecture outperforms the existing methods in terms of the dice score while also having the lowest standard deviation, which depicts its robustness against the variations in the type and size of lung nodules. In comparison with our previous work [5], which utilized the cropped ROI input, our current approach offers improved performance with a lower standard deviation. This can be attributed to the incorporation of ROI masks into a dual-encoder-based architecture, which eliminates the necessity to crop and normalize the input slice. It also signifies the effectiveness of the incorporation of the A-ROI algorithm in the proposed scheme to estimate the ROI masks for the surrounding slices of a given input slice.

### 5.2. Robustness Analysis

The LIDC dataset includes annotations that describe various characteristics of nodules, such as their subtlety, internal structure, calcification, sphericity, margin, lobulation, speculation, texture, and malignancy. These characteristics represent different levels of difficulty in detecting the boundaries of nodules. To evaluate the effectiveness of our method, we divided the test data into groups based on each characteristic and analyzed the results for each group. Table 2 presents the dice scores for each group, which demonstrates that our framework performs consistently in each group, and promising results can be obtained on all types of lung nodules. This can be attributed to the hard attention mechanism, which enables the proposed *DEHA*-Net to only focus on the given ROI region while leveraging the surrounding information to distinguish the nodule.

Further, to illustrate the robustness of the proposed method, a histogram of the distribution of the dice scores on the test set of the LIDC dataset is shown in Figure 5. The majority of test instances have a score of over 85%, which demonstrates the strong performance of our proposed method.

### 5.3. Qualitative Analysis

To elaborate on the difference in the performances of this framework and our previous work [5], we performed a visual analysis of the results. Figure 6 shows the visual results with axial views on randomly selected nodules of different sizes and types. The results demonstrate that the incorporation of hard attention with the ROI mask in the model significantly improves the segmentation performance. It can also be observed that the resizing of cropped slice disturbs the boundary of the segmented nodule, which is critical in determining the exact dimensions of the nodule and subsequently, the malignancy level. The proposed *DEHA*-Net enables our framework to utilize the full slice without losing minor details of the given input image, which are crucial to perform the accurate segmentation of lung nodules.

## 6. Conclusions

In this work, we proposed a novel dual-stage-based framework that used a 2D slice along with the seed region of interest (ROI), covering the nodule area, to produce the 3D segmentation of the nodule. To segment the nodule in the given slice, we proposed a novel dual-encoder-based hard attention network (*DEHA*-Net), which utilized the adaptive region of interest (A-ROI) algorithm for estimating the ROI for the surrounding slices. In contrast to the previous studies in which a cropped patch of a given slice is inputted to the network, the proposed *DEHA*-Net leverages complete 2D contextual information by taking the entire slice as input. It helps the *DEHA*-Net to learn the meaningful features that better distinguish between the nodule and non-nodular voxels. In the second stage, after obtaining the 3D segmentation of nodules from axial slices, the framework followed the same segmentation scheme for sagittal and coronal views. Finally, a consensus module was employed to process the results from all three axes to obtain the refined segmentation mask. An extensive evaluation of the proposed framework was performed on the lung image database consortium and image database resource initiative (LIDC/IDRI) dataset, which is the largest publicly available dataset. The quantitative and qualitative results were presented and analyzed, which demonstrate that the technique shows excellent performance by outperforming the existing state-of-the-art methods in terms of the dice similarity score. Furthermore, our results reveal that the framework is significantly robust to the various types and sizes of nodules. Future plans include improvement in the framework by reducing its computational complexity to optimize its performance in terms of execution time.

## Figures and Tables

**Figure 1 sensors-23-01989-f001:**
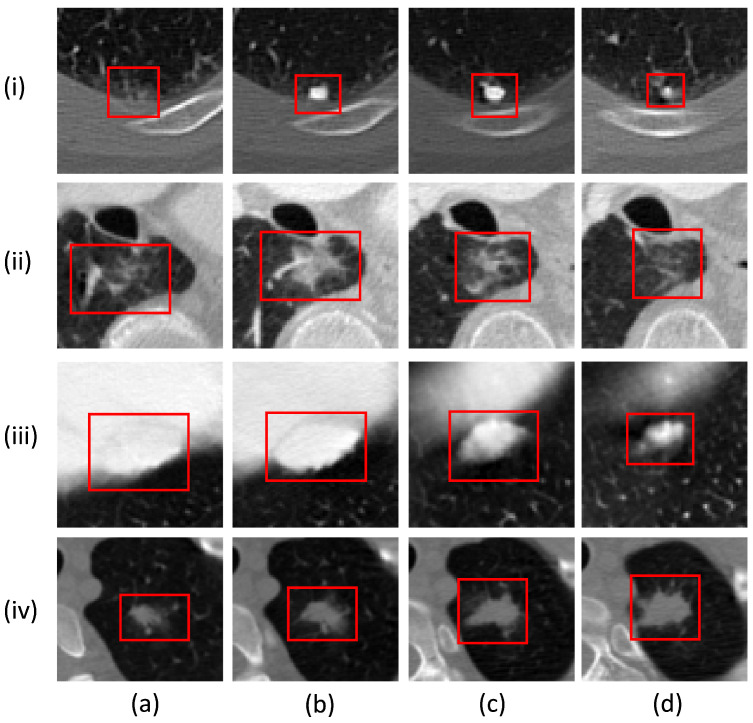
Illustration of different types of lung nodules. The intra-nodule diversity can be noticed in columns (**a**–**d**), whereas inter-nodule variations are demonstrated in rows (i–iv).

**Figure 2 sensors-23-01989-f002:**
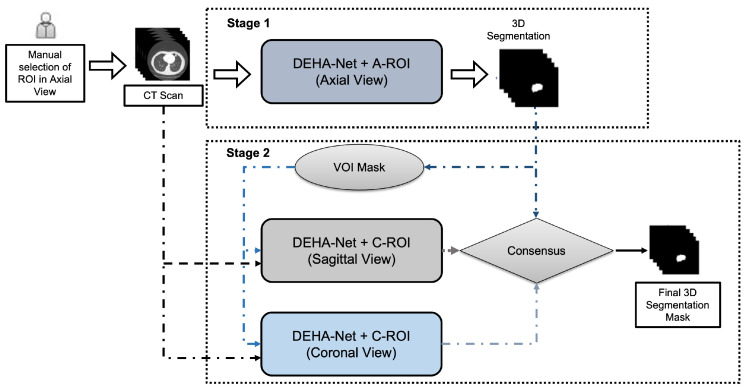
The proposed framework consists of two stages. In the first stage, the user or CADe system provides the ROI along the axial axis, and the *DEHA*-Net (dual-encoder-based hard attention network with self-hard attention) and adaptive ROI algorithm are used to determine the ROIs in the surrounding slices to perform 3D segmentation. In the second stage, the sagittal and coronal views are created to segment the nodule. Finally, three segmentation predictions are fed into the consensus module to produce the final 3D segmentation mask.

**Figure 3 sensors-23-01989-f003:**
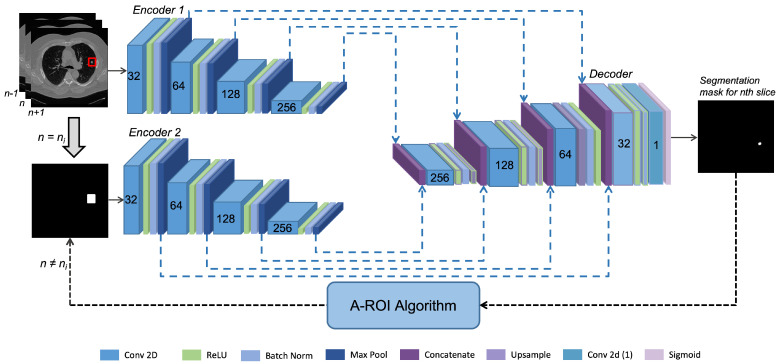
Illustration of the proposed dual–encoder–based hard attention network (*DEHA*-Net) architecture, which consists of two encoder blocks and one decoder block to incorporate the hard attention for lung nodule segmentation.

**Figure 4 sensors-23-01989-f004:**

Constant and adaptive regions of interest (ROIs) are depicted in a series of consecutive slices containing a nodule from column (**a**–**f**). Te segmentation of the nodule starts from column (**a**) with the manual ROI and ends at column (**f**). The blue and red bounding boxes represent constant ROIs, while the green boxes represent adaptive ROIs. (Figure credit: Usman et al. [5]).

**Figure 5 sensors-23-01989-f005:**
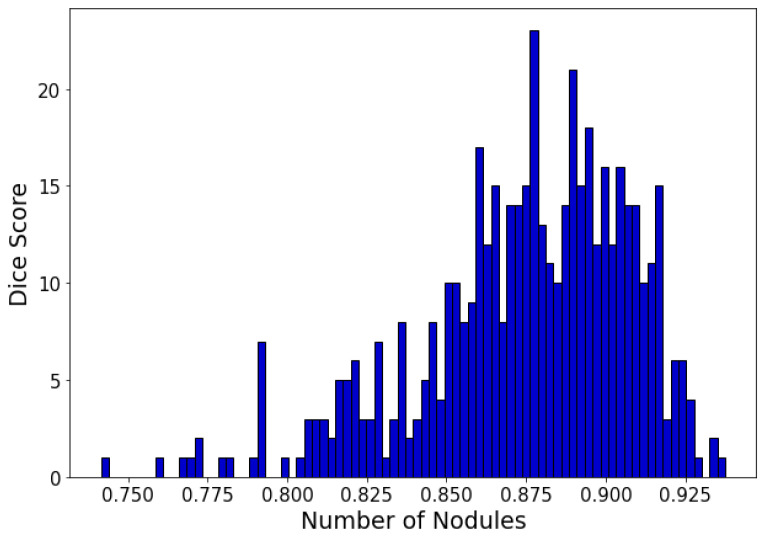
Dice similarity score distribution obtained on the LIDC testing set.

**Figure 6 sensors-23-01989-f006:**
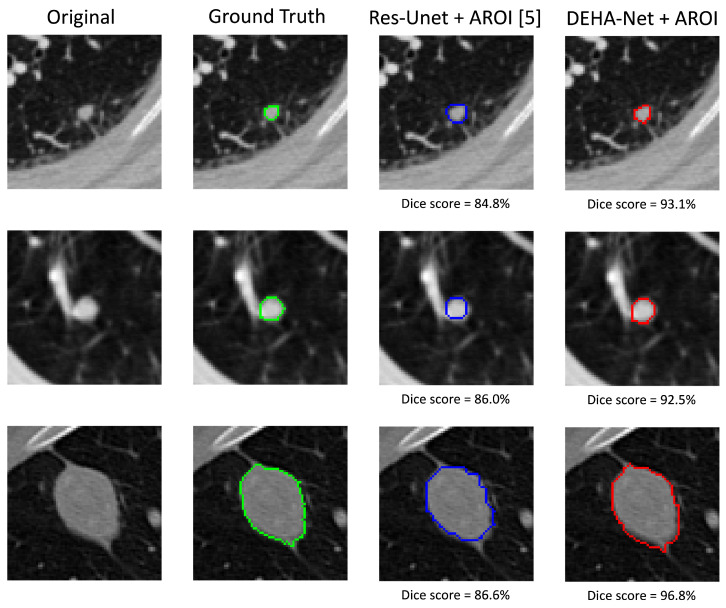
The visual results comparison of the previous cropped-slice-based approach with Res-UNet and full-slice input-based approach with *DEHA*-Net.

**Table 1 sensors-23-01989-t001:** The quantitative results of our proposed scheme along with previously published studies in terms of mean ± standard deviation of all quantitative measures used in this study. The best performance is indicated in bold, and *–* indicates the absence of value.

Authors, Year	DSC (%)	SEN (%)	PPV (%)
Wang et al., 2017 [19]	82.15 ± 10.76	92.75 ± 12.83	75.84 ± 13.14
Tong et al., 2018 [21]	73.6 ± *–*	*–*	*–*
Liu et al., 2019 [24]	81.58 ± 11.05	87.30 ± 14.30	79.71 ± 13.59
Chen et al., 2020 [23]	86.43 ± *–*	*–*	*–*
Cao et al., 2020 [37]	82.74 ± 10.20	89.35 ± 11.79	79.64 ± 13.34
Usman et al., 2020 [5]	87.55 ± 10.58	91.62 ± 8.47	88.24 ± 9.52
Chen et al., 2021 [25]	81.32 ± *–*	92.33 ± *–*	74.78 ± *–*
Maqsood et al., 2021 [38]	81 ± *–*	*–*	*–*
Zhang et al., 2022 [27]	85.1 ± 7.10	82.7 ± 10.8	**90 ± 10.7**
Tyagi et al., 2022 [18]	80.74 ± *–*	85.46 ± *–*	80.56 ± *–*
Chen et al., 2022 [25]	81.32 ± *–*	**92.33 ± –**	74.78 ± *–*
Zhou et al., 2022 [39]	86.75 ± 10.58	89.07 ± 8.31	83.26 ± 10.21
Our Method 2023	**87.91 ± 6.27**	90.84 ± 8.22	89.56 ± 10.07

**Table 2 sensors-23-01989-t002:** Mean dice score for various types of nodules from the LIDC-IDRI testing set.

Characteristics	Characteristic Score					
	1	2	3	4	5	6
Calcification	-	-	85.99 [18]	91.25 [42]	85.98 [27]	87.77 [405]
Internal structure	87.98 [487]	78.04 [3]	-	84.13 [2]	-	-
Lobulation	91.07 [201]	86.09 [164]	84.79 [78]	85.08 [31]	87.54 [18]	-
Malignancy	89.18 [39]	87.76 [114]	79.45 [163]	89.14 [98]	91.02 [78]	-
Margin	92.08 [9]	89.81 [37]	79.25 [78]	82.99 [232]	92.97 [136]	-
Sphericity		88.77 [38]	83.22 [153]	91.61 [218]	90.24 [83]	-
Speculation	92.42 [257]	82.69 [165]	85.17 [32]	80.39 [14]	83.56 [24]	-
Subtlety	80.3 [4]	88.96 [22]	82.88 [131]	91.99 [238]	86.03 [97]	-
Texture	80.47 [11]	85.73 [18]	87.1 [26]	82.27 [107]	90.17 [330]	-

## Data Availability

The data used in this study is public dataset from The Lung Image Database Consortium (LIDC) and Image Database Resource Initiative (IDRI) which can be accessed from https://wiki.cancerimagingarchive.net/pages/viewpage.action?pageId=1966254 (accessed on 5 February 2023). The architecture source code can be provided on request.

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
