# Peer review of "DEHA-Net: A Dual-Encoder-Based Hard Attention Network with an Adaptive ROI Mechanism for Lung Nodule Segmentation"

_sensors, 2023, doi:10.3390/s23041989_

Round 1

Reviewer 1 Report

The authors proposed the manuscript titled " DEHA-Net: Dual-Encoder based Hard Attention Network with Adaptive ROI Mechanism for Lung Nodule Segmentation ". A detailed description of the proposed work has been presented in the manuscript with results, however, the abstract is lacking the proper flow of the proposed work, results can be included to make it more useful.

Author Response

We would like to firstly thank the reviewer for his/her valuable feedback, which helped us in improving the quality of the paper in several places. We strived to incorporate each comment into consideration and did the necessary modifications in the revised version.

The reviewer comments requiring the author’s attention/ or requiring any changes are reproduced below, along with our response to each comment.

Reviewer: 1

Comment: The authors proposed the manuscript titled " DEHA-Net: Dual-Encoder based Hard Attention Network with Adaptive ROI Mechanism for Lung Nodule Segmentation ". A detailed description of the proposed work has been presented in the manuscript with results, however, the abstract is lacking the proper flow of the proposed work, results can be included to make it more useful.

Authors' Response: The authors express their sincerest gratitude for the time and effort you put into reviewing our manuscript. We have revised the abstract to improve the flow of the proposed framework and also included the results in the abstract.

Reviewer 2 Report

In the manuscript "DEHA-Net: Dual-Encoder based Hard Attention Network with Adaptive ROI Mechanism for Lung Nodule Segmentation" , the authors present a new method about medical image segmenting, especially design a dual-encoder-based hard attention network (DEHA-Net) , which exploits the adaptive region of interest to automatically investigate the penetration of lung nodule into surrounding slices. Comparisons have been made, further more its advantages and limitations have been discussed. However, there are still some concerns should be revised before it can be accepted:

1. Visualizations of the lung nodule have been shown in Figure 1, but it is not rigorous enough, it should be columns A to D for the article.

2. In Section 3.2, the functions should be described in more detail, such as using grayscale thresholds to exclude interference outside the CT value range contained in lung nodules, reducing the influence of noise and irrelevant tissues, normalizing the image can reduce the network's dependence on parameter initialization, and reduce the difficulty of the optimization process, so that the optimization process is significantly smoother and easier to converge to the optimal solution.

3. Illustration of proposed dual-encoder-based hard attention network (DEHA-Net), as shown in Figure 3, the number of convolution kernels is best marked on the structure to facilitate the reader's understanding.

4. When describing the encoder network structure in section 3.3.1, line 218 original text is five layers make a single level of encoder should be four layers make a single level of encoder.

5. In the article, the writing format has 3-D and 3D, which should be unified and standardized.

6. For Table 1, it may be more convincing to reproduce all the works done by the previous works mentioned in Section II.

In addition, the manuscript contains a number of spelling errors that could have been easily removed by using a spell checker. For instance, there is a spell error vois in line 41

Author Response

We wanted to take a moment to express our sincerest gratitude to the reviewer for the time and effort he/she put into reviewing our manuscript. The valuable feedback helped us in improving the quality of the paper in several places. We strived to incorporate each comment into consideration and did the necessary modifications in the revised version.

In this document, we provide a summary of all the modifications and point-by-point replies to the reviewer's recommendations. To improve the readability of this document, and to help differentiate the reviewer’s comments from our response, we have used different formatting in this document.

Bold: the reviewer comments;

Italic: the authors’ response to the reviewer's comments;

Reviewer 2 Comments:

The authors In the manuscript "DEHA-Net: Dual-Encoder based Hard Attention Network with Adaptive ROI Mechanism for Lung Nodule Segmentation" , the authors present a new method about medical image segmenting, especially design a dual-encoder-based hard attention network (DEHA-Net) , which exploits the adaptive region of interest to automatically investigate the penetration of lung nodule into surrounding slices. Comparisons have been made, further more its advantages and limitations have been discussed. However, there are still some concerns should be revised before it can be accepted:

  1. Visualizations of the lung nodule have been shown in Figure 1, but it is not rigorous enough, it should be columns A to D for the article.

The authors are grateful to the reviewer for providing an in-depth analysis and pointing out the typos. We have revised Figure 1 and corrected the mistake in the caption.

  1. In Section 3.2, the functions should be described in more detail, such as using grayscale thresholds to exclude interference outside the CT value range contained in lung nodules, reducing the influence of noise and irrelevant tissues, normalizing the image can reduce the network's dependence on parameter initialization, and reduce the difficulty of the optimization process, so that the optimization process is significantly smoother and easier to converge to the optimal solution.

The authors are grateful to the reviewer for providing the suggestion to improve the preprocessing section. As suggested, in the updated manuscript we have added more details about preprocessing in 3.2 section.

  1. Illustration of proposed dual-encoder-based hard attention network (DEHA-Net), as shown in Figure 3, the number of convolution kernels is best marked on the structure to facilitate the reader's understanding.

We appreciate the reviewer’s suggestion to improve the readability of Figure 3. In the updated manuscript, the number of kernels utilized at each layer has been mentioned in the figure as well.

  1. When describing the encoder network structure in section 3.3.1, line 218 original text is five layers make a single level of encoder should be four layers make a single level of encoder.

The authors thank the reviewer for highlighting the critical typo. The mistake has been fixed in the revised version.

  1. In the article, the writing format has 3-D and 3D, which should be unified and standardized.

The authors are grateful to the reviewer for the detailed analysis of the manuscript and for highlighting the inconsistencies. We have worked diligently to remove all the inconsistencies from the updated manuscript.

  1. For Table 1, it may be more convincing to reproduce all the works done by the previous works mentioned in Section II.

The authors appreciate the reviewer’s suggestion. All the previous deep learning-based studies mentioned in related work (Section II) have been included in comparison Table 1.

In addition, the manuscript contains a number of spelling errors that could have been easily removed by using a spell checker. For instance, there is a spell error vois in line 41

 We highly appreciate the reviewer’s efforts to identify the spelling mistakes. The manuscript has been thoroughly reviewed and the spelling mistakes have been fixed. We have also worked diligently on improving the paper’s readability and flow of arguments. It is hoped that the reviewers will find that the paper is in much better shape than before and all their comments have been satisfactorily addressed.

Round 2

Reviewer 2 Report

The article has been made more detailed modification, basically meet the requirements of publication